# Multiparametric Deep Learning Tissue Signatures for Muscular Dystrophy: Preliminary Results

**Alex E. Bocchieri**[3]                                                                      ABOCCHI2@JHU.EDU
**Vishwa S. Parekh**[1,3]                                                              VISHWAPAREKH@JHU.EDU
**Kathryn R. Wagner**[4]                                                    WAGNERK@KENNEDYKRIEGER.ORG
**Shivani Ahlawat**[1]                                                                    SAHLAWA1@JHMI.EDU
**Vladimir Braverman**[3]                                                                    VOVA@CS.JHU.EDU
**Doris G. Leung**[4]                                                                       DLEUNG8@JHMI.EDU
**Michael A. Jacobs**[1,2]                                                                   MIKEJ@MRI.JHU.EDU

[1] *The Russell H. Morgan Department of Radiology and Radiological Sciences, The Johns Hopkins University School of Medicine, Baltimore, MD 21205, USA*
[2] *Sidney Kimmel Comprehensive Center, The Johns Hopkins University School of Medicine, Baltimore, MD 21205, USA*
[3] *Department of Computer Science, The Johns Hopkins University, Baltimore, MD 21218*
[4] *Kennedy Krieger Center, Baltimore, MD 21205*

## Abstract

A current clinical challenge is identifying limb girdle muscular dystrophy 2I (LGMD2I) tissue changes in the thighs, in particular, separating fat, fat-infiltrated muscle, and muscle tissue. Deep learning algorithms have the ability to learn different features by using the inherent tissue contrasts from multiparametric magnetic resonance imaging (mpMRI). To that end, we developed a novel multiparametric deep learning network (MPDL) tissue signature model based on mpMRI and applied it to LGMD2I. We demonstrate that the new tissue signature model of muscular dystrophy with the MPDL algorithm segments different tissue types with excellent results.

**Keywords:** Deep learning, Machine learning, CNN, Magnetic resonance imaging, Multiparametric MRI, Muscular dystrophy, Tissue signature vector

## 1. Introduction

Deep learning algorithms are beginning to emerge in radiological applications for segmentation and classification of different diseases (Krizhevsky et al., 2012), (Ronneberger et al., 2015), (Litjens et al., 2017). These deep learning algorithms have the ability to learn different features by using the inherent tissue contrasts from multiparametric magnetic resonance imaging (mpMRI). To that end, we developed a novel multiparametric deep learning network (MPDL) tissue signature model based on mpMRI and applied to limb girdle muscular dystrophy 2I (LGMD2I) (Parekh et al., 2018). LGMD2I typically affects the muscles in the

shoulder and pelvic girdle (Wicklund and Kissel, 2014). We focused on muscle groups in the thighs, in which separating fat, fat-infiltrated muscle, and muscle tissue can be challenging.

## 2. Methods

We tested the MPDL segmentation with a total cohort of 30 subjects (19 patients and 11 normal subjects) that underwent whole body MRI (WB-MRI) (Leung et al., 2015). All studies were performed under the institutional guidelines for clinical research under a protocol approved by the Johns Hopkins University School of Medicine Institutional Review Board (IRB) and all HIPAA agreements. Whole body-MRI was performed at 3T with a Siemens scanner. WB-MRI sequences were obtained in the axial and coronal planes with continuous table movement (CTM) consisting of of T1, T2, Dixon, and diffusion weighted images. Figure 1 shows a typical WB-MRI and the resulting MPDL segmentation. Apparent Diffusion Coefficient (ADC) of water maps were constructed for quantitative analysis. The tissue signatures were defined for muscle, fat, fat-infiltrated tissue, and bone in the thighs from the Dixon images and used as inputs into a MPDL convolutional neural network (CNN) (Lecun et al., 1998), (Krizhevsky et al., 2012), (Ronneberger et al., 2015).

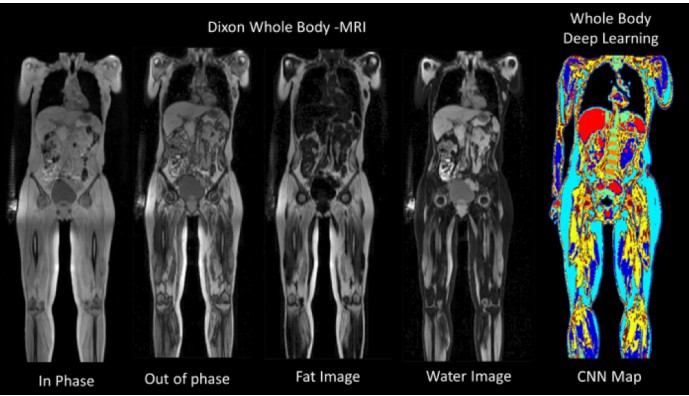

Figure 1: Example Dixon WB-MRI and typical MPDL segmentation map.

### 2.1. Multiparametric Deep Learning Tissue Signatures

The MPDL network builds a composite feature representation using the muscular dystrophy tissue signatures defined by a tissue signature vector as gray level intensity values corresponding to each voxel position within the images. Mathematically, the MPDL tissue signatures are defined as follows:

$MPDL\ Tissue\ Signatures = TS^{(\tau)} = [T_1^{(\tau)}, T_2^{(\tau)}, DIXON_n^{(\tau)}, ..., DWI_n^{(\tau)}]^T$

where $(\tau)$ is the tissue type and $n$ is the number of the images in the sequence. We defined a tissue signature for each of the following: background, normal muscle, normal fat and fat infiltrated tissue, bone, and skin. The MPDL network was trained on the muscular dystrophy tissue signatures defined from the MR images and shown in Figure 2.

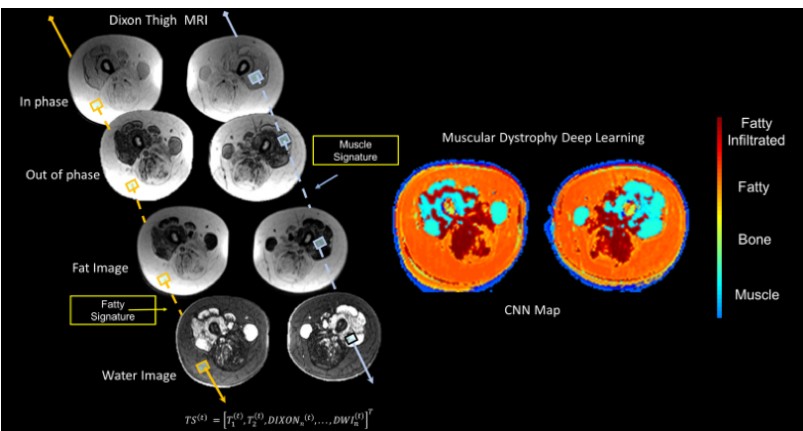

Figure 2: Demonstration of the MPDL tissue signature model and CNN segmentation map.

## 2.2. Convolutional Neural Network

The patch-based CNN was implemented and trained on the tissue signature image patches of size 5x5xN corresponding to each N dimensional tissue signature. The 5x5 image patch of a tissue signature corresponds to the immediate 5x5 neighborhood of that voxel position. The 2D-CNN consisted of four layers with 128, 64, 32 and 16 filters respectively, followed by a fully connected layer and a softmax layer (Nair and Hinton, 2010) and shown in Figure 3.

## 2.3. Evaluation

To compare the MPDL segmentation, we used the Eigenimage (EI) algorithm, which is a semi-supervised segmentation method to determine muscle and fatty tissue from the Dixon MRI and the MPDL signatures (Windham et al., 1988). Based on the segmentations of the different tissue types, we derived tissue fractions for each of the major tissue types of muscle, fatty, fatty infiltrated, and bone in both cohorts of subjects. The Dice Similarity (DS) metric was used for the overlap evaluation of the segmented regions (Dice, 1945). We calculated the DS metrics and ADC map values from segmented tissue types. A board-certified MSK radiologist confirmed the qualitative verification of the segmented regions. Regions of Interests (ROI) were drawn on the normal muscle and fat infiltrated tissue on the ADC map images for quantitative imaging metrics. Student's two-tailed t-test was used to determine any statistical significance between the ADC values of muscle and fat infiltrated tissue. We set statistical significance at $p < 0.05$.

## 3. Results

In this study there were a total of 30 subjects. Nineteen LGMD2I patients (12 females and seven male ($37 \pm 13$ years old)) and 11 normal volunteers. The Dice metrics of the MPDL segmentation in normal volunteers were $0.92 \pm 0.04$ for fat and $0.88 \pm 0.03$ for muscle. In patients, the Dice metrics were $0.81 \pm 0.09$ for fat and $0.82 \pm 0.04$ using MPDL CNN. There were significant differences between normal and fat-infiltrated muscle ADC map values

(Right Side: $1.46 \pm 0.21$ versus $0.92 \pm 0.21 \times 10^{-3}$ mm$^2$/s and Left Side: $1.45 \pm 0.24$ versus $0.88 \pm 0.20 \times 10^{-3}$ mm$^2$/s). There were no differences between each normal thigh muscle ADC values. The CNN segmentation maps of muscle, fat and fat infiltrated tissue was qualitatively confirmed by the radiologist.

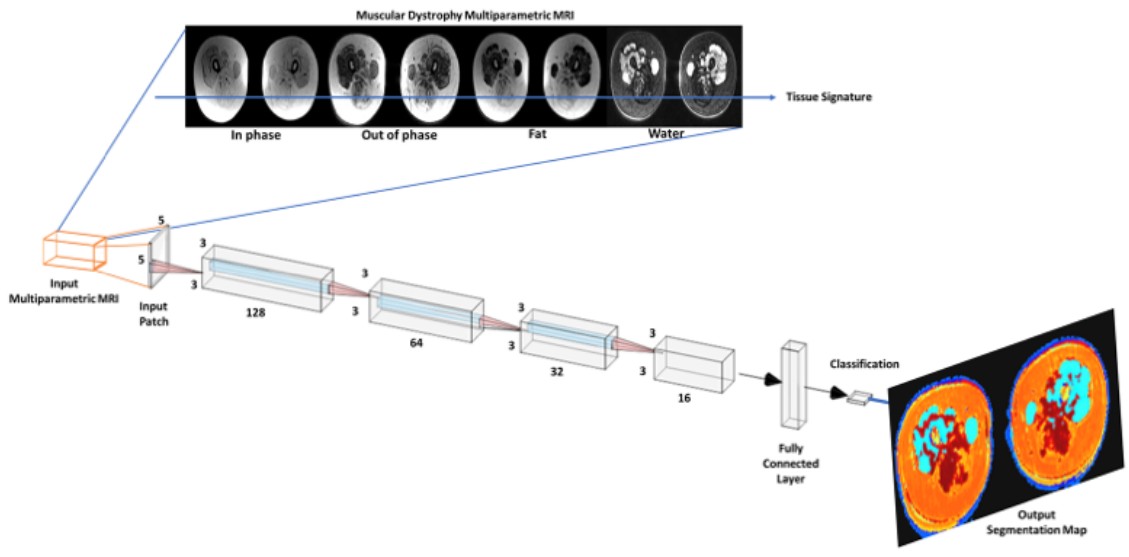

Figure 3: The CNN architecture for the MPDL model.

## 4. Conclusion

We have introduced a new tissue signature model of fat and muscle in LGMD2I muscular dystrophy patients that can be used as inputs into an MPDL network algorithm. The application of the MPDL tissue signature model resulted in excellent segmentation and classification of different tissue types in muscle and fat infiltration of muscle in muscular dystrophy. We could use these heterogeneous regions within the muscle for further classification of the degrees of muscle replacement by fat in muscular dystrophy using quantitative ADC maps. There are ongoing research studies to compare the MPDL CNN results to Stacked Sparse Autoencoders and U-Net DLN. These initial results need to be validated in a larger cohort of subjects and more specifically to the present study, any assessment of the clinical value of MPDL network will require additional studies in a larger patient population with a prospective trial with subsequent follow-up and clinical correlation using MPDL. This would provide us with new data to explore the exact application and methods to apply to larger studies. In conclusion, we have demonstrated that integrated MPDL methods accurately segmented and classified different muscular dystrophy tissue from multiparametric muscular dystrophy MRI.

## Acknowledgments

National Institutes of Health (NIH) grant numbers: 5P30CA006973 (Imaging Response Assessment Team - IRAT), U01CA140204, 1R01CA190299, and the Tesla K40s used for this research were donated by the NVIDIA Corporation.

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
