# OpenReview forum: "Multiparametric Deep Learning Tissue Signatures for Muscular Dystrophy: Preliminary Results"
_MIDL.io/2019/Conference/Abstract — MIDL Abstract 2019_

### Official Review · AnonReviewer2 · 2019-04-30
**comparison to more recent works**

**Rating:** 3
**Confidence:** 2

**Review:**

The authors proposed MPDL segmentation with MPDL tissue signatures to segment  muscle, fat and fat infiltrated tissues and reported Dice scores on dataset from the thighs of whole body MRI scans and the results agreed with the evaluation from radiologists.
The results can be more sound if the authors compare it to more recent and state-of-the-art works of segmentation.

---

### Official Review · AnonReviewer1 · 2019-05-01
**promising and important application; found evaluation/results a little confusing**

**Rating:** 3
**Confidence:** 1

**Review:**

The authors propose an approach for segmenting tissue types based on a “tissue signature model”, in which multi-parametric MR images are combined as input to a CNN. High performance was achieved.
* The clinical application to muscular dystrophy research is important, and (at least to my knowledge) the approach is novel.
* I found the Evaluation and Results sections a bit confusing. The SSAE map (Figure 2) wasn't described.
* The Methods mention that the proposed approach was compared to an Eigenimage algorithm, but unless I missed it, I didn’t see results of the EI segmentation reported.

---

### Decision · Program_Chairs · 2019-05-06
**Acceptance Decision**

Accept